# Movement Disorders in Children with a Mitochondrial Disease: A Cross-Sectional Survey from the Nationwide Italian Collaborative Network of Mitochondrial Diseases

**DOI:** 10.3390/jcm10102063

**Published:** 2021-05-12

**Authors:** Chiara Ticci, Daniele Orsucci, Anna Ardissone, Luca Bello, Enrico Bertini, Irene Bonato, Claudio Bruno, Valerio Carelli, Daria Diodato, Stefano Doccini, Maria Alice Donati, Claudia Dosi, Massimiliano Filosto, Chiara Fiorillo, Chiara La Morgia, Costanza Lamperti, Silvia Marchet, Diego Martinelli, Carlo Minetti, Maurizio Moggio, Tiziana Enrica Mongini, Vincenzo Montano, Isabella Moroni, Olimpia Musumeci, Elia Pancheri, Elena Pegoraro, Guido Primiano, Elena Procopio, Anna Rubegni, Roberta Scalise, Monica Sciacco, Serenella Servidei, Gabriele Siciliano, Costanza Simoncini, Deborah Tolomeo, Paola Tonin, Antonio Toscano, Flavia Tubili, Michelangelo Mancuso, Roberta Battini, Filippo Maria Santorelli

**Affiliations:** 1IRCCS Fondazione Stella Maris, 56018 Pisa, Italy; chiara.ticci@fsm.unipi.it (C.T.); stefano.doccini@fsm.unipi.it (S.D.); dosiclaudia@gmail.com (C.D.); anna.rubegni@fsm.unipi.it (A.R.); roberta.scalise@fsm.unipi.it (R.S.); deborah.tolomeo@fsm.unipi.it (D.T.); filippo3364@gmail.com (F.M.S.); 2Unit of Neurology, San Luca Hospital, 55100 Lucca, Italy; daniele.orsucci@uslnordovest.toscana.it; 3Child Neurology, Fondazione IRCCS Istituto Neurologico Carlo Besta, 20133 Milan, Italy; Anna.Ardissone@istituto-besta.it (A.A.); isabella.moroni@istituto-besta.it (I.M.); 4Neuromuscular Unit, Department of Neuroscience, University of Padova, 35121 Padua, Italy; luca.bello@unipd.it (L.B.); elena.pegoraro@unipd.it (E.P.); 5Bambino Gesù Children’s Hospital IRCCS, 00165 Rome, Italy; ebertini@gmail.com (E.B.); daria.diodato@opbg.net (D.D.); diego.martinelli@opbg.net (D.M.); 6Center of Translational and Experimental Myology, IRCCS Istituto Giannina Gaslini, 16147 Genova, Italy; ire.bonato92@gmail.com (I.B.); claudio2246@gmail.com (C.B.); 7Programma di Neurogenetica, IRCCS Istituto delle Scienze Neurologiche di Bologna, 40139 Bologna, Italy; valerio.carelli@unibo.it (V.C.); chiara.lamorgia@unibo.it (C.L.M.); 8Department of Biomedical and Neuromotor Sciences (DIBINEM), University of Bologna, 40126 Bologna, Italy; 9A. Meyer Children Hospital, 50139 Florence, Italy; maria.donati@meyer.it (M.A.D.); flavia.tubili@meyer.it (F.T.); elena.procopio@meyer.it (E.P.); 10Department of Clinical and Experimental Sciences, University of Brescia, NeMO-Brescia Clinical Center for Neuromuscular Diseases, 25064 Brescia, Italy; massimiliano.filosto@unibs.it; 11Neuromuscular Disorders Unit, IRCCS Istituto Giannina Gaslini, DINOGMI, University of Genoa, 16147 Genoa, Italy; chi.fiorillo@gmail.com (C.F.); minettic@unige.it (C.M.); 12IRCCS Istituto delle Scienze Neurologiche di Bologna, UOC Clinica Neurologica, 40139 Bologna, Italy; 13Genetics and Neurogetics, Fondazione IRCCS Istituto Neurologico C. Besta, 20133 Milan, Italy; costanza.lamperti@istituto-besta.it (C.L.); maerchet.s@istituto-besta.it (S.M.); 14Neuromuscular and Rare Diseases Unit, Fondazione IRCCS Ca’ Granda Ospedale Maggiore Policlinico, Dino Ferrari Centre University of Milan, 20122 Milan, Italy; mmmoggio@gmail.com (M.M.); monica.sciacco@policlinico.mi.it (M.S.); 15Neuromuscular Unit, Department of Neuroscience “Rita Levi Montalcini”, University of Torino, 10124 Torino, Italy; tmongini@cittadellasalute.to.it; 16Department of Clinical and Experimental Medicine, Neurological Institute, University of Pisa, 56126 Pisa, Italy; v.montano89@gmail.com (V.M.); gsicilia@neuro.med.unipi.it (G.S.); costanza.simoncini85@gmail.com (C.S.); mancusomichelangelo@gmail.com (M.M.); 17Unit of Neurology and Neuromuscular Disorders, Department of Clinical and Experimental Medicine, University of Messina, 98125 Messina, Italy; omusumeci@unime.it (O.M.); atoscano@unime.it (A.T.); 18Neurological Clinic, University of Verona, 37134 Verona, Italy; eliapancheri88@gmail.com (E.P.); paola.tonin@ospedaleuniverona.it (P.T.); 19Fondazione Policlinico Universitario A. Gemelli IRCCS, 00168 Rome, Italy; guido.primiano@gmail.com (G.P.); serenella.servidei@policlinicogemelli.it (S.S.); 20Dipartimento Universitario di Neuroscienze, Università Cattolica del Sacro Cuore, 00168 Roma, Italy; 21Tuscan PhD Program of Neuroscience, University of Florence, Pisa and Siena, 50139 Florence, Italy

**Keywords:** mitochondrial disease, movement disorder, childhood onset, multicenter cross-sectional study

## Abstract

Movement disorders are increasingly being recognized as a manifestation of childhood-onset mitochondrial diseases (MDs). However, the spectrum and characteristics of these conditions have not been studied in detail in the context of a well-defined cohort of patients. We retrospectively explored a cohort of individuals with childhood-onset MDs querying the Nationwide Italian Collaborative Network of Mitochondrial Diseases database. Using a customized online questionnaire, we attempted to collect data from the subgroup of patients with movement disorders. Complete information was available for 102 patients. Movement disorder was the presenting feature of MD in 45 individuals, with a mean age at onset of 11 years. Ataxia was the most common movement disorder at onset, followed by dystonia, tremor, hypokinetic disorders, chorea, and myoclonus. During the disease course, most patients (67.7%) encountered a worsening of their movement disorder. Basal ganglia involvement, cerebral white matter changes, and cerebellar atrophy were the most commonly associated neuroradiological patterns. Forty-one patients harbored point mutations in the mitochondrial DNA, 10 carried mitochondrial DNA rearrangements, and 41 cases presented mutations in nuclear-DNA-encoded genes, the latter being associated with an earlier onset and a higher impairment in activities of daily living. Among our patients, 32 individuals received pharmacological treatment; clonazepam and oral baclofen were the most commonly used drugs, whereas levodopa and intrathecal baclofen administration were the most effective. A better delineation of the movement disorders phenotypes starting in childhood may improve our diagnostic workup in MDs, fine tuning management, and treatment of affected patients.

## 1. Introduction

Movement disorders have become an increasingly recognized neurological manifestation of mitochondrial diseases (MDs), which can present clinically in isolation, combined with other movement disorders, or they can be part of a multisystemic presentation [1]. Even when isolated, these should in some cases prompt a search for a MD; indeed, the diagnosis of MD is crucial for genetic counseling, and should lead to care for any associated multisystemic disorders [2].

Even though massive gene testing has expanded the phenotypes associated with MD, the characteristics of movement disorder in childhood-onset MDs remain largely undefined and need to be further clarified. Most reports consist of a single case description or small case series [3,4,5,6,7], or retrospective analysis conducted mainly on adult patients with different inclusion criteria [8,9]. No multi-center registry-based studies have systematically delineated the spectrum or progression of mitochondrial movement disorders in the context of a clinically and genetically defined cohort of patients, especially in forms with childhood-onset.

More specifically, little is known about the type of movement disorders, both at onset and during the course of MDs, and about their associated neuroradiological, molecular, and therapeutic features. Because of the wide phenotypic variability seen in MDs, identification of the underlying genetic basis of these movement disorders is critical to better understand the molecular pathophysiology and to develop effective therapeutic targets.

The aim of the present study was to better investigate the clinical, neuroradiological, and genetic features of movement disorders in childhood-onset MDs on the basis of data collected through the Nationwide Italian Collaborative Network of Mitochondrial Diseases (NICNMD) [10,11,12,13,14,15,16,17,18].

Better delineation of the phenotypes in MDs starting in childhood may help to guide diagnosis and follow-up of affected individuals, facilitate timely interventions, and contribute to the development of genetic testing pipelines.

## 2. Materials and Methods

We retrospectively explored a cohort of individuals with childhood-onset MDs (<16 years) querying the NICNMD database. The use of the database was approved by the ethics committees of all the centers involved in the study. These centers obtained written informed consent from all participating patients or their legal guardians, in accordance with the ethical standards of the 1964 Declaration of Helsinki. Patients are included in the NICNMD dataset only if their diagnosis of MD is supported by consistent clinical, histological, biochemical, and molecular findings. All the centers in the network have specific expertise in MDs, both in children and in adults. Patients have been included in the database since 2010 until 1 September 2019. At the time of this study, the NICNMD contains cross-sectional information of 1467 Italian patients.

The clinical section of the dataset includes dichotomous (“yes or no”) items that allowed us to subdivide the whole sample into two groups, according to the presence or absence of movement disorders, and to analyze the main clinical and molecular features of each of these two groups. Subsequently, by means of a customized online questionnaire sent by email to all the NICNMD centers, we investigated the main clinical and genetic features in the cohort of patients with movement disorders. We included patients with ataxia and hypokinetic disorders, as well as cases showing predominant hyperkinetic features including chorea, myoclonus, dystonia, tremor, paroxysmal dyskinesia, and tics [19].

Data on age at onset of movement disorder, type of manifestations both at onset and during the course of MD, associated comorbidities, pharmacological treatments, and neuroradiological and molecular findings were collected automatically in a downloadable online spreadsheet. Survey questions on the various types of movement disorders were created using the latest and most shared literature classifications. In particular, dystonia was classified according to body distribution (focal, segmental, multifocal, generalized, and hemidystonia) and temporal pattern (persistent, action-specific, diurnal fluctuations, and paroxysmal) as reported elsewhere [20]. Myoclonus description was based on body distribution (focal, segmental, multifocal, and generalized) and provoking factors (spontaneous, action, and reflex) [21]. Tremor was classified according to anatomical distribution (focal, segmental, hemitremor, and generalized) and activation conditions (rest-tremor and action-tremor; the latter includes postural, kinetic (simple and intention), task-specific, and isometric tremors) [22]. Associated clinical features (e.g., bradykinesia, stiffness, tremor, and hypomimia) were collected for hypokinetic disorders [23]. In addition, ataxia was distinguished in cerebellar, sensory, and spinocerebellar disorders; associated clinical signs (e.g., dysmetria, nystagmus, and adiadokinesia) and body sites and functions involved (trunk, walking, and gesture, etc.) were also taken into consideration [24]. We subdivided the entire set of patients according to the type of movement disorder occurring at onset into three main subgroups, namely, hyperkinetic (chorea, dystonia, myoclonus, tics, paroxysmal dyskinesia, and tremor), hypokinetic, and ataxic onset.

In order to evaluate the course of movement disorder over time, we set the online questionnaire to collect information on the motor skills of patients and their level of dependence in activities of daily living, assessed both at the onset and at the last follow-up. We also collected data on standardized rating scales used to measure the severity of movement disorder [25] if available, and reported the evolution over time and whether the course of the movement disorder was stable, or fluctuations were seen. We also asked centers (all experts in evaluating and treating patients with movement disorders) to cite drugs used in their patients and to report dichotomously their expert global impression of change since last assessment (using a “yes” or “no” response to specific treatment).

Regarding the neuroradiological features, we considered atrophy of brain structures as a brain parenchymal volume loss over longitudinal scans. Therefore, we asked all centers to report possible cerebellar, cerebral, and brainstem atrophy based on qualitative assessments in at least two structural MR studies (average time between the two exams was 18 months on average). In this study, we did not request or analyze quantitative data.

Genotypes were divided into two main subgroups: those related to mtDNA mutations and nDNA mutations, and into seven more specific subcategories: m.3243A > G (the so-called common MELAS mutation), m.8344A > G (the so-called common MERRF mutation), m.8993T > G (usually associated with neuropathy, ataxia, retinitis pigmentosa (NARP), and maternally inherited Leigh’s syndrome (MILS)), m.10197G > A (*MT-ND3* mutation), additional mtDNA point mutations, mtDNA rearrangements, and the whole group of nDNA mutations. Statistical analyses were conducted using the IBM© SPSS© Statistics software, version 20 (Armonk, New York, NY, USA) and MedCalc^®^ software, version 18.10.2 (MedCalc Software Ltd., Ostendm Belgium). Fisher’s exact test was used for categorical associations. Bonferroni’s correction for multiple tests was applied where appropriate. The normally distributed continuous variables were compared by unpaired two-tailed Student’s t-test. Variables with abnormal distribution were analyzed using the nonparametric Mann–Whitney U test. Statistical significance was set at a two-tailed *p* value of <0.05.

## 3. Results

### 3.1. Features of MD Patients with and without Movement Disorders

In a total of 1467 patients presented in the NICNMD registry, 580 had a fully reported clinical picture and a childhood onset (<16 years). Of these, 197 (34.0%) were reported as suffering from different movement disorders. The “movement disorder” group had mean age at onset of 3.8 ± 4.8 years and age at last evaluation was 17.6 ± 16.4 years, whereas disease duration was 13.8 ± 14.1 years. The “non-movement disorder” group consisted of 383 patients who had a mean age at onset of 6.9 ± 5.6 years; age at last evaluation was 26.0 ± 19.2 years; and disease duration was 19.1 ± 17.1 years. Age at disease onset was statistically significantly lower in the “movement disorder” group (*p* < 0.001). As expected, because of the earlier onset, age at last evaluation was also lower (*p* < 0.001), as well as disease duration *p* < 0.001). Males were predominant for the “movement disorder” group 105/197 (53.3%, M/F 1.14) but this did not differ significantly from what we observed in the other group (185/383, 48.3%, M/F 0.93).

Table 1 shows the prevalence of the most frequently associated clinical features in patients with movement disorders. Most of these features were more common in the “movement disorder” group (i.e., generalized myopathy, cognitive involvement, pyramidal signs, hearing loss, failure to thrive, and short stature).

Results of neuroimaging could be re-examined in 180/197 and in 211/383 patients with and without movement disorders, respectively. MRI scans were abnormal in 165/180 (91.7%) patients with movement disorders, and in 126/211 (59.7%) of the other individuals (*p* < 0.001). Therefore, it appears that movement disorders were more frequently linked to a brain structural damage.

### 3.2. Google Survey Results on MD Patients with Movement Disorders

#### 3.2.1. Clinical, Neuroradiological, and Genetic Features

Through the online questionnaire, we obtained information on the main clinical, neuroradiological, and genetic features in 102/197 (51.8%) patients showing movement disorders and MD starting in childhood (Figure 1; Table 2 and Table 3). Of the patients, 58 were males and 44 were females, with a mean age at MD onset of 4.6 ± 4.5 years. The duration of follow-up ranged from 1 to 47 years (mean 11.1 ± 10.1 years).

In 45 of the 102 individuals collected in the multicenter survey (44.1%), the movement disorder was the presenting feature of MD, whereas in the other 56 subjects (54.9%) movement disorder was preceded by other mitochondrial symptoms (Figure 1A). Sensory organ dysfunction, ophtalmoparesis/ptosis and neuromuscular disorders were the most commonly reported mitochondrial comorbidities of the movement disorder (Figure 1B).

Brain magnetic resonance data in 98 patients presented a normal imaging in 6 (6.1%); basal ganglia involvement, cerebral white matter changes, and cerebellar atrophy were the most commonly reported neuroradiological patterns (Table 2).

The age at onset of movement disorder ranged from 0 to 70 years (mean 11.0 ± 13.3 years), with a childhood onset (<16 years) in 74/102 (72.5%) affected individuals. Ataxia was the most common movement disorder at onset (50; 49%), followed by dystonia (18; 17%), tremor (14; 14%), hypokinetic disorders (9; 9%), chorea (5; 5%), myoclonus (4; 4%), and one case each in paroxysmal dyskinesia and tics. Table 4 details the main features at onset of each type of movement disorder and their median age at onset. During the course of the disease, 40 patients (39.2%) suffered from a type of movement disorder different from that they presented at onset, with a significantly increased risk in individuals in whom movement disorder was the presenting symptom of MD (65% vs. 35% of patients in whom movement disorder was preceded by other comorbidities; *p* = 0.001).

During the MD course, the movement disorder worsened in most patients (69/102; 67.7%), remained stable in 25 (24.5%), and improved in only in a minority of patients (8/102; 7.8%). The worsening of the movement disorder was in line with the worsening of their motor functions and of the level of dependence in activities of daily living, evaluated at the last follow-up as a comparison with those seen at onset (Figure 2).

Scores of standardized rating scales used to measure the severity of movement disorder were available in only 7 patients (4 had an assessment of SARA, Scale for the Assessment and Rating of Ataxia [26]; 3 with the MD-CRS, Movement Disorder-Childhood Rating Scale [27]).

A total of 33/102 individuals (26%) received pharmacological treatment for the movement disorder: a monotherapy was adopted in 57.6% (19/33) and a polytherapy in 42.4% (14/33) (Figure 3). Multiple vitamins and cofactors, referred to as the “mitochondrial cocktail” [28], were used in 91/102 patients (89.2%), with a reported efficacy in modulating movement disorder in 13.

Forty-one patients (44.6%) harbored point mutations in the mitochondrial DNA, 10 (10.9%) had mtDNA rearrangements (9/10 single deletions; 1/10 multiple deletions), and 41 (44.6%) carried mutations in nuclear-DNA-encoded genes (Table 3). Age at movement disorder onset was found to differ significantly between individuals harboring nDNA mutations (median: 2 years) and those with mtDNA mutations (median: 10 years) (*p* = 0.001). Moreover, individuals with nDNA mutations, as opposed to those with mtDNA mutations, had a significantly higher probability of displaying partial or complete dependence in activities of daily living at the last follow-up (84.6% vs. 57.1%, *p* = 0.006).

#### 3.2.2. Clinical, Neuroradiological, and Genetic Features According to the Type of Movement Disorder at Onset

##### Hyperkinetic Onset Subgroup

Patients with hyperkinetic movement disorders at onset (43/102; 42.2%) had a mean age at MD onset of 3.9 ± 4.8 years. Hyperkinetic movement disorder was the presenting feature of MD in 20 (46.5%) individuals, with onset at a mean age of 6.3 ± 9.1 years. Dystonia and chorea have an earlier onset compared with other hyperkinetic movement disorders (Table 4). Sensory organ dysfunction and neuromuscular disorder were the main clinical features going before movement disorder and sensory organ dysfunction. Gastrointestinal manifestation and failure to thrive were the main comorbidities (Figure 1). During the course of the disease, 13/43 (30.2%) patients suffered from a type of movement disorder different from the disorder that they presented at onset (8 individuals with tremor at onset, and 5 with dystonia). In patients with tremor at onset, ataxia was the most frequent manifestation during the course (5/8), followed by dystonia (2/8), and ataxia or myoclonus (1/8). During the MD course, dystonia was associated with chorea or ataxia each in 2 patients, and hypokinetic disorder in a single case. The most frequent brain imaging abnormalities were basal ganglia involvement and cerebral white-matter changes (Table 2). Nuclear DNA mutations were the most frequent alteration of this subgroup (see Table 3). Of note 17/43 individuals (39.5%) received pharmacological treatment for the movement disorder. Oral baclofen, clonazepam, and trihexyphenidyl were the most commonly used drugs; levetiracetam and intrathecal baclofen were the most effective (the former used in patient with myoclonus, the latter in patients with chorea and dystonia) (Figure 3).

##### Hypokinetic Onset Subgroup

Nine patients had hypokinetic movement disorder at onset (8.8%) with a mean age at MD onset of 5 ± 4.1 years; the age at onset of movement disorder ranged from 0 years to 70 years (mean 19.4 ± 24.4 years). Ophtalmoparesis/ptosis was the main comorbid feature occurring before or accompanying movement disorder (Figure 1). During the course of the disease, hypokinetic movement disorder was followed by a different type of movement disorder in 3/9 (33.3%) of patients (dystonia, tremor, and ataxia each in one case); all three patients had a hypokinetic movement disorder onset of less than 10 years. Brain magnetic resonance imaging most often revealed basal ganglia involvement and cerebral white-matter changes (Table 2). Most patients harbored mutations in nuclear genes (see Table 3). Levodopa was the most commonly used and effective drug (Figure 3).

##### Ataxic Onset Subgroup

In patients with ataxia as presenting movement disorder (50/102; 49.0%), MD started at a mean age of 5.2 ± 4.4 years. Ataxia was the first feature of MD in about half of the individuals (21/50; 42.0%), with onset at a mean age of 13.5 ± 12.6 years. Sensory organ dysfunction, ophtalmoparesis/ptosis, and epilepsy were the main clinical features going before ataxia. Sensory organ dysfunction, ophtalmoparesis/ptosis, and neuropathy were the main comorbidities (Figure 1).

During the course of the disease, 17/50 (34%) patients suffered from a different type of movement disorder. Among them, 7 had dystonia, 5 hypokinetic disorder, 1 showed dystonia and tremor, 1 dystonia combined with chorea, and a single case only myoclonus. Dystonia was focal in 5/9 individuals, multifocal in 2/9, and generalized in 2/9. According to the temporal pattern, dystonia was action-specific in 5, persistent in 3 and paroxysmal in a single patient. Cerebellar atrophy, basal ganglia involvement, and cerebral white matter changes were the most commonly reported neuroradiological patterns (Table 2). As in other subgroups, nuclear DNA mutations were the most common genetic data (Table 3). Ten individuals (10/50; 20%) received pharmacological treatment for the movement disorder; clonazepam was the most commonly used drug (Figure 3).

## 4. Discussion

We here report the findings of a multicenter study providing a cross-sectional assessment of clinical and genetic features of movement disorders in MD starting in childhood. The NICNMD registry used in this study has already helped to redefine the clinical phenotype of common mtDNA mutations in Italian mitochondrial patients (e.g., m.3243A > G [12], m.8344A > G [13], or single deletions [10]) and it has been used to investigate several features of MDs, including myoclonus [14], ocular myopathies [11], peripheral neuropathy [15], muscle pain [17], lipomatosis [18], and epilepsy [16]. As of the time of this writing, the registry contains complete information on 1467 patients (last accession date 1 September 2019).

Data on the prevalence of movement disorders in the context of MD are limited and conflicting. Frequency rate in small studies was quantified retrospectively as 11% in biopsy-proven cases [4], whereas the numbers are different ranging from 6.2% to 30% in genetically or biochemically confirmed cases with movement disorders [8,9]. No study reports population-based cross-sectional data, and sampling and referral bias, heterogeneous age at onset in addition to the genetic heterogeneity of the cohorts, most likely explain this difference. Our large multi-center series of most genetically-confirmed cases is the first to be performed in childhood-onset only forms of MDs and it shows that movement disorders are frequent clinical manifestations of MDs. The relatively higher prevalence of movement disorders in our cohort (34%) may be partially related to the fact that we also included patients with ataxia in the movement disorder subgroup, as done by others [9].

In line with a previous paper reporting a high frequency of movement disorder as the presenting symptom of MD [9], we observed that movement disorder was the feature at onset in 44% of the sample in the Italian NICNMD database. This result underlines the importance of a more precise phenotypic description, aimed to guide clinicians towards the suspicion of a mitochondrial etiology in a movement disorder of unknown cause. Literature data on the type and detailed features of movement disorders presented at the onset and during the course of mitochondrial disease are scarce. In our study, we observed that a movement disorder appeared on average 5 years from the onset of MDs and ataxia was the most common presentation at onset (49%), followed by dystonia (17%), tremor (14%), hypokinetic disorders (9%), chorea (5%), myoclonus (4%), paroxystic dyskinesia (1%), and tics (1%). Dystonia and chorea had an earlier onset (median: 1 year) compared with other movement disorders, starting at the end of the first decade of life (Table 4). At the onset, dystonia was mainly generalized and persistent; chorea instead frequently affected the limbs function. Mitochondrial ataxia was frequently cerebellar at onset, with limbs and walking involvement and high frequency of dysmetria, dysarthria, and hyporeflexia or areflexia. Tremor was primarily segmental, whereas myoclonus generalized (Table 4).

During the course of the disease, the movement disorder could be slightly modified in 40% of the patients. In patients with tremor at onset, ataxia was the most frequent manifestation during the course, followed by dystonia and myoclonus ataxia. During the disease course, dystonia was associated with chorea, ataxia, and hypokinetic disorders, whereas hypokinetic disorders were followed by mixed features; also 34% of ataxic patients suffered from a different type of disorder, mostly dystonia. Dystonia in the disease course assumed different characteristics from those of the onset, as it mainly resulted in focal and action-specific features.

Overtime, the movement disorder worsened in most patients, with a concomitant worsening of the motor functions and the level of dependence in activities of daily living (Figure 2). Although the retrospective nature of this study prevents us from conclusions on natural history, overall these data add information to the scarcity of literature on mitochondrial movement disorders and may help clinicians to predict the possible type and course of these neurological conditions in MD.

Comparison of the features of MD patients with or without a history of movement disorder revealed, in the former, a significantly lower age at MD onset, higher rates of MRI abnormalities, and more common comorbidities (see Table 1). This result indicated that mitochondrial movement disorder depicts a more complex multisystem phenotype within the category of MD, with an earlier onset and prominent involvement of the central nervous system. As previously reported [8,9], the MRI abnormalities in our cohort were heterogeneous and included both supra- and sub-tentorial impairments and brainstem involvement (Table 2). Cerebellar atrophy was reported to be the predominant MRI finding in paediatric patients with respiratory chain defects [29]; in our cohort a high prevalence of cerebellar involvement is described not only in patients with ataxia at onset, but also in other subgroups. These findings underline the emerging role of cerebellar dysfunction in the pathogenesis of movement disorders, probably due to the cerebellum connection with basal ganglia, through a disynaptic reciprocal neuronal network [30,31,32,33]. Pharmacological therapies likely act as neurotransmitter modulators in this complex neuronal pathway, e.g., using dopamine and γ- aminobutyric acid (GABA) enhancing agents [34].

Our knowledge of MDs is rapidly growing as new genes are identified and new phenotypes described. Early diagnosis of MDs has become essential to provide appropriate genetic counseling, management, and treatment of affected patients. Proper recognition of an underlying mitochondrial defect in patients presenting with movement disorder is therefore of paramount importance. In half of our sample, movement disorder was not a presenting feature of MD but was preceded by other symptoms, of which the most frequent were eye and hearing dysfunction, ophtalmoparesis/ptosis, and epilepsy (Figure 1A). Comparison of the clinical features of MD patients with different types of movement disorder at onset (Figure 1) revealed, in individuals with hyperkinetic onset, a higher prevalence of neurodevelopmental regression or delay and hypotonia as clinical features preceding movement disorder onset, and a higher risk of failure to thrive and gastrointestinal involvement during the disease course. Cardiovascular disease and migraine were more frequent in patients with hypokinetic onset, whereas neuropathy mainly affected individuals with ataxia at onset. Taken together, these findings confirm that it is often the multisystemic clinical phenotype that leads to suspect a mitochondrial aetiology and suggest that all individuals with mitochondrial movement disorder deserve, as part of their follow-up, careful screening for this, or at least proper tests for eye and ear dysfunctions, assessment of muscle impairment, and electroencephalographic study. Moreover, the cardiovascular system should also be evaluated in cases presenting with hypokinetic movement disorder of the peripheral nervous system in those with ataxic onset. Conversely, it is important to assess swallowing and gastrointestinal dysfunction and to perform a correct nutritional management in children with hyperkinetic movement disorders at onset.

Despite the widespread use of massive parallel DNA sequencing in the current MD clinical diagnostic setting, little is known about genotype–phenotype correlations in mitochondrial movement disorders. Many studies have reported relatively small cohorts and included patients with genetically unclassified disease, thereby lacking sufficient detail to formulate meaningful correlations between the phenotype and underlying gene defect [35]. The clinical variability of MDs, and the consequent low statistical power of single-center studies, represents an additional factor which make it difficult to establish reliable correlations in this setting. In our study, 41 patients (44.6%) harbored point mutations in the mtDNA, 10 (10.9%) mtDNA rearrangements, and 41 (44.6%) carried mutations in nDNA-encoded genes (Table 3). Individuals harboring nDNA mutations, as opposed to those with mtDNA mutations, had a significantly lower age at movement disorder onset and a higher impairment in activities of daily living at the last follow-up. A predominant gene or a predominant mutation did not appear, reflecting the heterogeneous movement disorder manifestations; however, it appeared that children with hyperkinetic onset had a quite high occurrence of mutations in the *MT-ND3* gene [36]. Identification of a relatively high number of MERRF mutation carriers among patients with ataxic and hyperkinetic onset is not too surprising, since ataxia, myoclonus, and tremor were reported as frequent features in a retrospective database-based study on a large cohort of patients with the m.8344A **>** G mutation [13]. MtDNA rearrangements were also relatively more frequent in patients with ataxic and hyperkinetic onset; this finding is in line with the new proposed criteria for Kearns–Sayre Syndrome spectrum including ataxia and tremor as additional clinical features [10]. Identification of a relatively high number of MELAS and NARP mutation carriers among patients with ataxic onset represented a finding already reported by others [12,37] and explained the high frequency of neuropathy we recorded in the ataxic-subgroup. Nonetheless, the reason for this appealing association between a given mutation and some clinical features remains undefined.

Two final comments are worth mentioning as results of this work. Health-related quality of life (HRQOL) in children with inborn errors of metabolism and co-existent movement disorder is significantly reduced compared with other chronic, stigmatizing disorders; moreover, a more severe movement disorder has been associated with a lower HRQOL [38]. Therefore, an accurate description of movement disorders, particularly in the context of complex, multi-organ disorders like MDs, is likely to aid therapeutic symptomatic management. In our cohort of 102 individuals with MD, only 33 patients (26%) received pharmacological medication to specifically treat the movement disorder (Figure 3). Multiple vitamins and cofactors were used in many but with a reportedly low efficacy over the patients’ movement disorder (14.3%). The limited number of publications addressing MD and movement disorders, coupled with the small number of cases in this cohort and in other studies that had received specific treatment targeting their movement disorder, suggests there may be little evidence to guide decisions on medical therapies at present. In our cohort, levodopa (dopamine precursor used to replace dopamine in the neuronal circuits) was used in four patients (three with hypokinetic movement disorder, one with generalized dystonia) and was effective in all. Seven patients with hyperkinetic movement disorder at onset were treated with oral Baclofen (selective agonist of GABA B receptors), with a modest clinical response as already reported [8], whereas clonazepam (GABAergic agent) used in six patients with hyperkinetic-onset and in two patients with ataxia appeared more effective when patients did not display dystonia. In the subgroup of patients with hyperkinetic movement disorder at onset, levetiracetam (inhibitor of potassium and calcium channels and interferes with the release of neurotransmitters) and intrathecal baclofen were the most effective treatment (the former used in patients with myoclonus, the latter in individuals with chorea or dystonia). While the use of levetiracetam in mitochondrial myoclonus has already been reported [39], the efficacy of intrathecal baclofen is a new finding in the literature, and it might deserve future attention in larger longitudinal studies.

Overall, our study is essentially a clinical, cross-sectional investigation, although performed in only 102/197 (51.8%) of MD patients with movement disorders listed in the Italian NICNMD. The purely retrospective nature of the study with no longitudinal follow-up makes it impossible to draw any firm conclusions on the natural history of mitochondrial movement disorder, or to precisely define the responses to different treatments. Longitudinal studies are needed to further assess the factors able to predict the risk of movement disorder in MD patients, and at the same time, assess the clinical and neuroimaging features that might point to a mitochondrial cause of movement disorder in this population. One additional limitation of our study, also connected to the retrospective nature, is the scarcity of data related to the use of standardized scales to assess the disease course of the movement disorder. Although there are not yet effective disease-modifying therapies, an increasing number of pharmacological clinical trials are being conducted in MDs [40]. The use of sensitive and valid endpoints is essential to prepare well-stratified clinical cohorts to test the effectiveness of potential treatments.

## 5. Conclusions

This study presents further evidence of the success of data sharing and represents the first Italian multicenter survey of movement disorders in MDs. Taking advantage of a large unselected cohort of childhood-onset MD patients, we contributed to the description of the clinical, neuroradiological, and genetic phenotype related to movement disorders.

Our results might help to guide early diagnostic workup, management, and treatment of affected patients, while also contributing to the implementation of clinically meaningful endpoints in future interventional trials. The observations concerning the apparent beneficial effects of intrathecal baclofen in children deserve further exploration in larger studies.

## Figures and Tables

**Figure 1 jcm-10-02063-f001:**
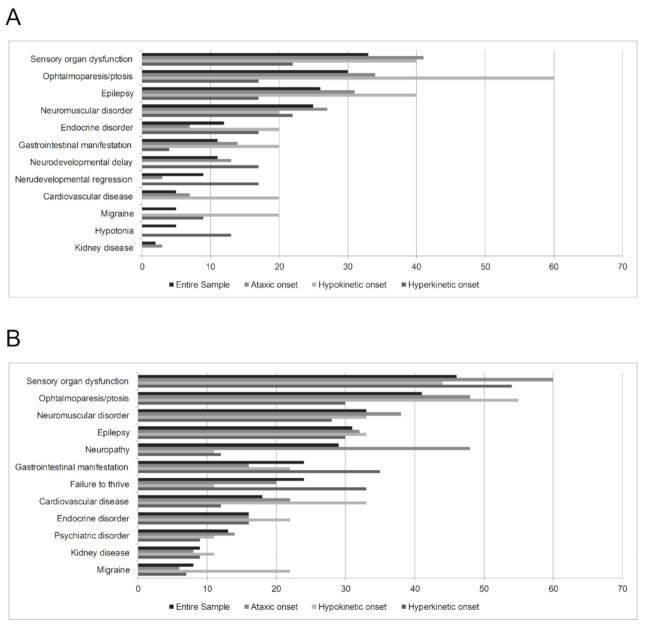
Symptoms preceding the movement disorder (in subjects in whom the movement disorder is not the first clinical feature; *n* = 56) (**A**) and comorbidities (**B**) in our cohort (*n* = 102), described both for the entire group and for the different movement disorder type-onset subgroups (values reported as percentages).

**Figure 2 jcm-10-02063-f002:**
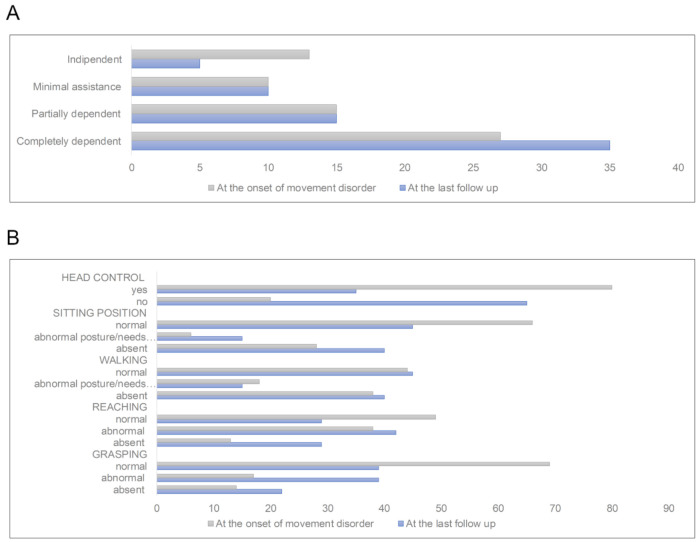
Level of dependence in activities of daily living and motor functions in mitochondrial patients with movement disorder. (**A**) Level of dependence in activities of daily living, assessed both at the onset of movement disorder and at the last follow-up. Data were available for 65 patients only. (**B**) Motor function assessed both at the onset of movement disorder and at the last follow-up. Data on onset motor skills were available for 64 patients; data on last follow-up skills were available for 58 patients. Values are reported as number of patients.

**Figure 3 jcm-10-02063-f003:**
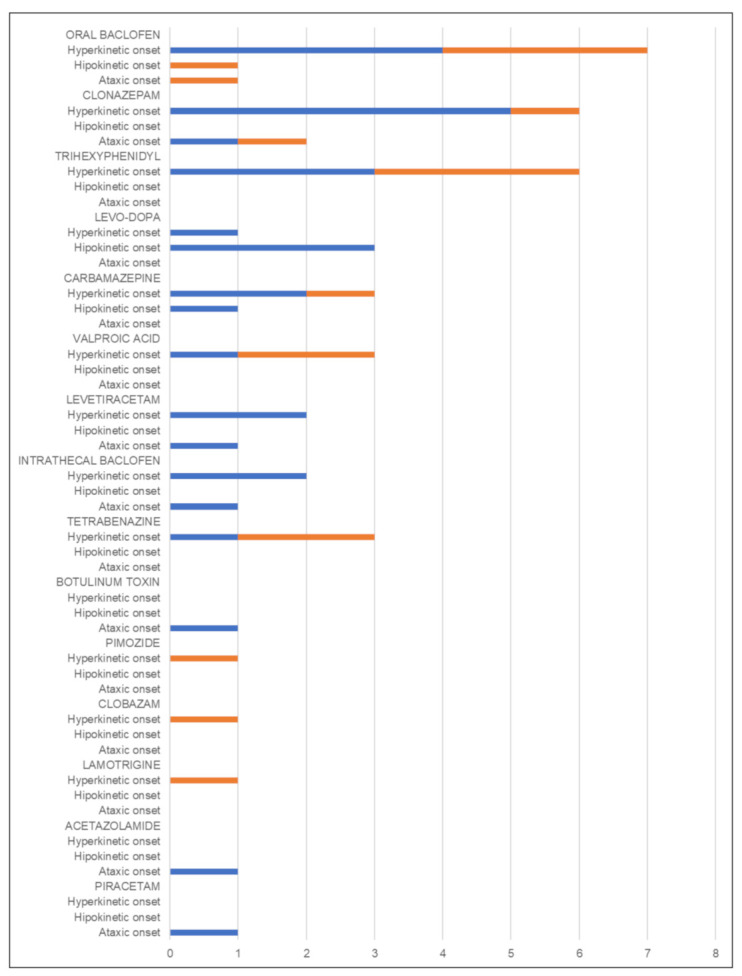
Use and efficacy of pharmacological treatments in patients with mitochondrial movement disorders. Values are reported as number of patients. Blue bars indicate the number of patients for whom the treatment was described as effective in modulating the movement disorder. Orange bars indicate the number of patients for whom the treatment was described as not effective in modulating the movement disorder.

**Table 1 jcm-10-02063-t001:** Phenotype-based findings in childhood-onset mitochondrial patients with and without movement disorders. Patients included in the database of the Nationwide Italian Collaborative Network of Mitochondrial Diseases were divided into two groups: with and without movement disorders. Clinical features present in fewer than 10% in the first group were not considered and are not shown. Ocular myopathy includes eyelid ptosis and ophthalmoparesis. Generalized myopathy includes at least one of the following: muscular weakness, exercise intolerance, muscle wasting, hypotonia, muscle pain, or myoglobinuria. Significance level after Bonferroni’s correction: 0.0033. Significant differences are shown in bold. n.s.: not significant.

Phenotype	Movement Disorders: Yes (*n* = 197)	Movement Disorders: No (*n* = 383)	Significance Level
**Generalized myopathy**	**151 (76.6%)**	**160 (41.8%)**	***p*** **< 0.001**
**Cognitive involvement**	**103 (52.3%)**	**64 (16.7%)**	***p*** **< 0.001**
Ocular myopathy	88 (44.7%)	147 (38.4%)	n.s.
**Pyramidal involvement**	**78 (39.6%)**	**39 (10.2%)**	***p*** **< 0.001**
**Hearing loss**	**73 (37.1%)**	**58 (15.1%)**	***p*** **< 0.001**
**Failure to thrive and short stature**	**70 (35.5%)**	**55 (14.4%)**	***p*** **< 0.001**
**Epileptic seizures**	**58 (29.4%)**	**49 (12.8%)**	***p*** **< 0.001**
**Peripheral neuropathy**	**43 (21.8%)**	**19 (5.0%)**	***p*** **< 0.001**
**Swallowing impairment**	**38 (19.3%)**	**31 (8.1%)**	***p*** **< 0.001**
**Retinopathy**	**34 (17.3%)**	**20 (5.2%)**	***p*** **< 0.001**
**Migraine**	**34 (17.3%)**	**31 (8.1%)**	***p*** **= 0.001**
Gastrointestinal dysmotility/vomiting	31 (15.7%)	33 (8.6%)	n.s.
**Respiratory involvement**	**29 (14.7%)**	**25 (6.5%)**	***p*** **= 0.002**
Cardiac involvement	27 (13.7%)	41 (10.7%)	n.s.
**Optic neuropathy**	**26 (13.2%)**	**133 (34.7%)**	***p*** **< 0.001**

**Table 2 jcm-10-02063-t002:** MRI abnormalities seen in mitochondrial paediatric-onset patients with movement disorders. *n* = 98 (imaging data were not available for 4 patients). MRI was completely normal in 6 patients (6.1%), with no significant difference between the subgroups. The association between ataxic onset and cerebellar atrophy is the only significant association in this table, with a border-line significance (significance level after Bonferroni’s correction: 0.002).

	Entire Sample (*n* = 98)	Hyperkinetic Onset (*n* = 41)	Hypokinetic Onset (*n* = 9)	Ataxic Onset (*n* = 48)
**Basal ganglia abnormalities**	53 (54.1%)	24 (58.5%)	6 (66.7%)	23 (47.9%)
**Cerebral white matter abnormalities**	47 (48.0%)	20 (48.8%)	6 (66.7%)	21 (43.8%)
**Cerebellar atrophy**	**41 (41.8%)**	**10 (24.4%)**	**3 (33.3%)**	**28 (58.3%)**
				***p*** **= 0.002**
**Cerebral atrophy**	34 (34.7%)	15 (36.6%)	3 (33.3%)	16 (33.3%)
**Cerebellar white matter abnormalities**	19 (19.4%)	6 (14.6%)	1 (11.1%)	12 (25.0%)
**Brainstem atrophy**	11 (11.2%)	5 (12.2%)	2 (22.2%)	4 (8.3%)
**Thalamic/subthalamic** **involvement**	13 (13.3%)	5 (12.2%)	1 (11.1%)	7 (14.6%)
**Dentate nucleus alterations**	11 (11.2%)	5 (12.2%)	2 (22.2%)	4 (8.3%)
**Stroke-like lesions**	5 (5.1%)	1 (2.4%)	2 (22.2%)	2 (4.2%)
**Cortical laminar necrosis**	4 (4.1%)	2 (4.9%)	1 (11.1%)	1 (2.1%)

**Table 3 jcm-10-02063-t003:** Genetic findings underlying mitochondrial movement disorders. *n* = 92 (genetic data were unknown for 10 patients). No significant differences were observed between the subgroups (significance level after Bonferroni’s correction: 0.002).

MUTATION	Entire Sample (*n* = 92)	Hyperkinetic Onset (*n* = 39)	Hypokinetic Onset (*n* = 9)	Ataxic Onset (*n* = 44)
Nuclear DNA mutations	41 (44.6%)	17 (43.6%)	3 (33.3%)	21 (47.7%)
mtDNA rearrangements	10 (10.9%)	5 (12.8%)	2 (22.2%)	3 (6.8%)
m.8993T > G	8 (8.7%)	1 (2.6%)	0 (0.0%)	7 (15.9%)
m.8344A > G	6 (6.5%)	4 (10.3%)	0 (0.0%)	2 (4.5%)
m.3243A > G	6 (6.5%)	1 (2.6%)	1 (11.1%)	4 (9.1%)
MT-ND3 mutations	6 (6.5%)	5 (12.8%)	1 (11.1%)	0 (0.0%)
Additional mtDNA mutations	15 (16.3%)	6 (15.4%)	2 (22.2%)	7 (15.9%)

**Table 4 jcm-10-02063-t004:** Clinical features at onset of each type of movement disorder and their median age at onset.

**Ataxia** (10 years)	Subtypes cerebellar (32/50), spino-cerebellar (16/50), sensory (7/50)
	Body sites / functions involved trunk (18/50), limbs (33/50), walking (39/50)
	Features associated dysmetria 35/50, dysarthria (26/50), adiadokinesia (16/50), nystagmus (9/50), hyporeflexia/areflexia (20/50), hyperreflexia (2/50), hypotonia (15/50), hypertonia (3/50)
**Dystonia** (1 year)	Body distribution focal (2/18), multifocal (4/18), generalized (10/18), hemidystonia (2/18)
	Temporal pattern persistent (16/18), action specific (1/18), paroxysmal (1/18)
**Tremor** (8 years)	Body distribution focal (5/14), segmental (7/14), generalized (2/14)
	Activation conditions rest-tremor (3/14), postural tremor (4/14), simple kinetic tremor (2/14), intention tremor (3/14), task specific tremor (2/14)
**Hypokinetic disorder** (10 years)	Features associated bradykinesia (9/9), stiffness (6/9), hypomimia (3/9), postural instability (2/9)
**Chorea** (1 year)	Body distribution limbs (5/5), trunk (1/5), face (1/5
**Myoclonus** (11 years)	Body distribution multifocal (1/4), generalized (3/4)

## Data Availability

Data Available in https://www.mitocon.it/registro-pazienti/.

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
