# Peer review of "Movement Disorders in Children with a Mitochondrial Disease: A Cross-Sectional Survey from the Nationwide Italian Collaborative Network of Mitochondrial Diseases"

_jcm, 2021, doi:10.3390/jcm10102063_

Round 1

Reviewer 1 Report

The authors reported the findings of a multicenter study providing a cross-sectional assessment of clinical and genetic features of movement disorders as a manifestation of childhood-onset mitochondrial diseases. They found ataxia was the most common movement disorder at onset, followed by dystonia, tremor, hypokinetic disorders, chorea and myoclonus, commonly associated with neuroradiological abnormalities. They also revealed that clonazepam and oral baclofen were the most commonly used drugs, whereas levodopa and intrathecal baclofen administration the most effective. This is an interesting and important study, but I only had a few minor comments:

1) The authors should mention the criteria of cerebellar, cerebral, and brainstem atrophy.

2) I wonder how the authors determined the efficacy of pharmacological treatments in a uniform manner. The details should be described in the Materials and Methods.

3) In the Discussion, the authors proposed the involvement of cerebellar dysfunction in the pathogenesis of movement disorders. They should explain the mechanism by which levodopa and intrathecal baclofen administration were effective for the several types of movement disorders.

Author Response

Response 1

We thank this reviewer for his/her comments and we have added the definition of atrophy in the Method Section (lines 140-144), as requested. We retrospectively investigated main clinical, molecular and imaging features of the patients with movement disorders adopting an ad-hoc prepared online questionnaire distributed to all the centers of the Italian network (NICNMD). Having considered atrophy of brain structures as a brain parenchymal volume loss over longitudinal scans, we asked all Centers to report possible cerebellar, cerebral, and brainstem atrophy based on qualitative assessment in at least two structural MR studies (average time between the two exams was 18 months on average). In this study, we did not request or analyze quantitative data.

Response 2

We thank the expert referee for calling our attention to this point. In our survey, we asked Centers (all expert in evaluating and treating patients with movement disorders) to cite drugs used in their patients and to report dichotomously their expert global impression of change since last assessment (using “yes” or “no” response to specific treatment). This is better described in the revised version, Methods Section (lines 137-139). We attempted in few cases (less than 10) to collect data with rating scales assessing severity of movement disorders. However, the low number we collected did not allow to draw meaningful conclusions (see also the Discussion Section, lines 518-530).

Response 3

We thank the reviewer for his/her comment. This is a difficult point to answer as the mechanisms of action of drugs in mitochondrial movement disorders, and their effectiveness, represent a still unexplored field, and perhaps our speculations go beyond the aim of this work. As reported in the original manuscript the role of cerebellar dysfunction in the pathogenesis of movement disorders is probably due to its connections with basal ganglia (see lines 446-447), through a disynaptic reciprocal neuronal network (see PMID: 23422326). Pharmacological therapies act likely as neurotransmitter modulators in this complex neuronal pathway, e.g. using dopamine and γ- aminobutyric acid (GABA) enhancing agents (PMID: 27302239). We added in the discussion a brief paragraph (lines 447-449); references list has been updated and renumbered accordingly. The hypothetic dysfunction of these pathways in children with movement disorders might propose the use of a dopamine replacement therapy for dystonia and hypokinetic movement disorders (see Figure 3).  Also, baclofen, a selective agonist of GABA-B receptors concentrated in the basal ganglia and the spinal cord, might enhance inhibitory neurotransmission. Intrathecal baclofen therapy is also being increasingly used in children with spasticity, generalized dystonia, myoclonus and cerebellar tremor (see PMID: 14567614; PMID: 27302239).  We added in the text the mechanisms of action of these drugs (lines: 506, 509, 510, 512,513).

We hope that in this new form the manuscript is acceptable for publication

Reviewer 2 Report

The study by Ticci et al. provides detailed description and overview of movement disorders in cohort of childhood-onset MD patients using "Nationwide Italian Collaborative Network of Mitochondrial Diseases" database. The aim of the study is to help and improve early diagnostics and implementation of clinically meaningful interventions. I found Manuscript to be little hard to follow, with a lot of repetition of the numbers already presented in the Tables.

The Introduction could be improved, and if there are similar studies from other countries it could be included as a reference.

There are some inconsistencies in the Tables:

The numbers don’t add up for:

Table 2 - Cerebral atrophy 32 (32.7%) 15 (36.6%) 3 (33.3%) 16 (33.3%)

Brainstem atrophy 13 (13.3%) 5 (12.2%) 2 (22.2%) 4 (8.3%)  and

Cortical laminar necrosis 3 (3.1%) 2 (4.9%) 1 (11.1%) 1 (2.1%).

Table 3 - mtDNA rearrangements 10 (10.9%) 5 (12.8%) 1 (11.1%) 3 (6.8%)

Additional mtDNA mutations 15 (16.3%) 6 (15.4%) 3 (33.3%) 7 (15.9%)

Please provide full name for ‘‘KSS spectrum’’ (line 473)

Author Response

Response 1

We thank the expert reviewer for his/her comment. We have improved the Introduction Section (lines 78-81), even using data taken from the Discussion ( from which they have been removed to avoid repetition, see lines 386-389) and adding comments to similar and earlier studies which were properly cited. References list has been updated and renumbered accordingly.

Response 2

We apologize for the mistakes. In the revised manuscript, we reviewed the data and corrected Tables 1 and 2. We also provided full name for ‘‘KSS spectrum’’, as suggested (line 488).

We hope that in this new form the manuscript is acceptable for publication

Reviewer 3 Report

The authors used a large national database "Nationwide Italian Collaborative Network of Mitochondrial Diseases" to investigate the clinical, neuro-radiological and genetic features of movement disorders in mitochondrial diseases. This database of 1467 patients, included between 2010 and 2019, was fully documented for 580 children. 197 of them, 34%, were suffering from movement disorders. Brain anatomical damage were observed for 92% of patients with movement disorders (165 out of 180 with neuroimaging available; 17 missing MRI) and 60% of patients without movement disorders (126 out of 211; 172 missing MRI).

Following phenotypes were higher for mitochondrial disease patients with movement disorders: generalized myopathy, cognitive involvement, pyramidal involvement, hearing loss failure to thrive and/or short stature, epileptic seizures, peripheral neuropathy, swallowing impairment, retinopathy, migraine and respiratory involvement.

Complete clinical, neuro-radiological and genetic features were obtained for 102 out of 197 mitochondrial diseases patients with movement disorders. Here would be my criticism of this work: it is a pity that the expert centers could not gather more data to respond to this online survey.

The presenting feature was movement disorders (ataxia in half of the cases) in 44% of patients and a secondary feature in others. The age of onset of studied disorders was 11 yo. During the course of mitochondrial disease, these movement disorders worsened in 68% of patients, evolving into new types of movement disorders in 39% of children. A specific corresponding pharmacological treatment was proposed for 26% of patients with movement disorders.

Author Response

We are grateful to this referee for his/her comment. This multi-center study was retrospective and could not get information from patients enrolled in the Italian network registry but lost at follow-up, whose information were therefore not retrievable and/or not initially reported in the database at the time of enrollment.

We have included this limitation in the revised manuscript (see lines 518-519). 

We hope that in this new form the manuscript is acceptable for publication

Round 2

Reviewer 2 Report

/